# NVRC: Neural Video Representation Compression

**Ho Man Kwan†, Ge Gao†, Fan Zhang†, Andrew Gower‡, David Bull†**
† Visual Information Lab, University of Bristol, UK
‡ Immersive Content & Comms Research, BT, UK
{hm.kwan, ge1.gao, fan.zhang, dave.bull}@bristol.ac.uk,
andrew.p.gower@bt.com

## Abstract

Recent advances in implicit neural representation (INR)-based video coding have demonstrated its potential to compete with both conventional and other learning-based approaches. With INR methods, a neural network is trained to overfit a video sequence, with its parameters compressed to obtain a compact representation of the video content. However, although promising results have been achieved, the best INR-based methods are still out-performed by the latest standard codecs, such as VVC VTM, partially due to the simple model compression techniques employed. In this paper, rather than focusing on representation architectures, which is a common focus in many existing works, we propose a novel INR-based video compression framework, Neural Video Representation Compression (NVRC)[1], targeting compression of the representation. Based on its novel quantization and entropy coding approaches, NVRC is the first framework capable of optimizing an INR-based video representation in a fully end-to-end manner for the rate-distortion trade-off. To further minimize the additional bitrate overhead introduced by the entropy models, NVRC also compresses all the network, quantization and entropy model parameters hierarchically. Our experiments show that NVRC outperforms many conventional and learning-based benchmark codecs, with a 23% average coding gain over VVC VTM (Random Access) on the UVG dataset, measured in PSNR. As far as we are aware, this is the first time an INR-based video codec achieving such performance.

## 1 Introduction

In recent years, learning-based video compression [40, 34, 36, 13, 30] has demonstrated its significant potential to compete with conventional video coding standards, with some recent contributions (e.g., DCVC-DC [36]) reported to outperform the latest MPEG standard codec, VVC VTM [10]. However, learning-based codecs are typically associated with high computational complexity, in particular at the decoder, which therefore limits their practical deployment. To address this, a new type of learned video codec has been proposed, based on implicit neural representation (INR) models [13, 30], where each INR instance is overfitted and compressed to represent a video sequence (or a video dataset). INR-based codecs enable much faster decoding speed compared to most non-INR learning based coding methods, and do not require offline optimization due to its overfitting nature. Although they have shown promise, INR-based codecs are yet to compete with state-of-the-art conventional and learned video coding methods in terms of rate-distortion performance.

To enhance coding performance, it is noted that most recent INR-based video coding methods [13, 30] focus on improving network architectures but still perform simply model pruning, quantization and entropy coding to obtain compact representations. Moreover, these methods are not fully end-to-end optimized; for example, NeRV [13] and HiNeRV [30] are not trained with a rate-distortion

---

[1]Project page: `https://hmkx.github.io/nvrc/`

38th Conference on Neural Information Processing Systems (NeurIPS 2024).

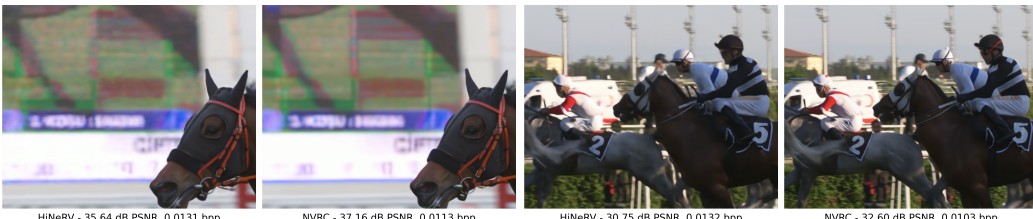

| HiNeRV - 35.64 dB PSNR, 0.0131 bpp | NVRC - 37.16 dB PSNR, 0.0113 bpp | HiNeRV - 30.75 dB PSNR, 0.0132 bpp | NVRC - 32.60 dB PSNR, 0.0103 bpp |

Figure 1: Comparison between the output from HiNeRV [30] and the proposed NVRC. The image is from UVG dataset (Jockey/ReadySetGo sequence) [45]

objective but only perform fine-tuning with pruning and quantization applied. Although COOL-CHIC [33] and C3 [29] are almost end-to-end optimized, the rate consumed by their entropy model and decoder/synthesis networks does not contribute to the training process. In contrast, state-of-the-art non-INR learning-based codecs [34, 35] are typically end-to-end trained with advanced entropy models, and this contributes to their improved coding performance compared to INR-based methods.

To address this issue, this paper proposes a new framework, referred to as Neural Video Representation Compression (NVRC). Unlike other INR-based video codecs, NVRC is an enhanced framework for compressing neural representations, which, for the first time, enables INR-based coding methods to be fully end-to-end optimized with advanced entropy models. In particular, NVRC groups network parameters and quantizes them with per-group learned quantization parameters. The feature grids are then encoded by a context-based entropy model, where the network layer parameters are compressed by a dual-axis conditional Gaussian model. The quantization and entropy model parameters are further compressed by a lightweight entropy model to reduce their bit rate consumption. The overall rate of the parameters from the INR, quantization, and entropy models is optimized together with the representation quality. NVRC also utilizes a refined training procedure, where the rate and distortion objectives are optimized alternatively to reduce the computational cost. The primary contributions of this work are summarized below.

1. The proposed NVRC is **the first fully end-to-end optimized INR-based framework** for video compression. In NVRC, neural representations, as well as quantization and entropy models, are optimized simultaneously based on a rate-distortion objective.

2. **Enhanced quantization and entropy models** have been applied to encode neural representation parameters, where the context and side information have been utilized to achieve higher coding efficiency.

3. A new **parameter coding method based on a hierarchical structure** has been introduced which allows NVRC to minimize the rate overhead. The parameters from quantization and entropy models for encoding the neural representation, are all quantized and coded with learnable parameters.

4. NVRC features an **enhanced training pipeline**, where the rate and distortion losses are optimized alternatively, to reduce the computational cost of advanced entropy models.

We conducted experiments to compare the proposed approach with state-of-the-art conventional and learning-based video codecs on the UVG [45], MCL-JCV [59] and JVET-CTC Class B [9] datasets. To enable a fair comparison, we use both the RGB444 (like most learned video codecs) and YUV420 (like standard video coding methods) configurations. The results demonstrate the effectiveness of NVRC, which achieved up to 23% and 50% BD-rate savings when compared to the latest MPEG standard codec, H.266/VVC VTM-20.0 (Random Access) [11], and the state-of-the-art INR-based codec, HiNeRV [30], respectively. To our best knowledge, NVRC is the **first INR-based video codec outperforming VVC VTM** with such significant coding gains.

## 2 Related work

### 2.1 Learning-based video compression

Video compression is an important research topic that underpins the development of many video-related applications, such as video streaming, video conferencing, surveillance, and gaming. In

the past three decades, multiple generations of video coding standards [60, 56, 10] have evolved by integrating advanced coding techniques. Recently, learning-based video compression emerged as an popular alternative due to its strong expressive power and the ability to be optimized in a data- and metric-driven manner. Neural networks can be combined with conventional codecs for performance enhancement [42, 64] or used to build end-to-end optimized frameworks. DVC [40] first proposed to replace all modules in conventional codecs using neural networks. Follow-up innovations include those improving motion estimation [38, 2, 26, 36], applying feature space conditional coding [27, 34], optimizing context modeling in terms of performance and efficiency [23, 22, 68], and adopting novel architectures such as normalizing flows [25], transformers [61, 44], *etc*. In addition to these architecture modifications, improvements to quantization-involved optimization have also been achieved to handle the non-differentiability caused by hard thresholding operations [29, 3, 19]. Moreover, several studies [17, 51, 28, 41, 39] have validated the effectiveness of adapting a model to an individual image of a video sequence via iterative refinement to reduce the amortization gap [63, 58] and optimize bit allocation over a sequence of frames [62]. Despite demonstrating impressive rate-distortion performance, with some recent advancements reporting outperformance of VVC [10], neural video codecs are generally too computationally intensive [48], thus limiting their adoption in practical applications.

### 2.2 INR-based video compression

Implicit neural representation (INR) [52] is an emerging paradigm for representing multimedia data, such as audios [55], images [14, 54], videos [13], and 3D scenes [46, 49]. This type of method exploits the mapping from the coordinate inputs to a high-dimensional feature space and aims to output the corresponding target data value at that location. Neural representation for videos (NeRV) [13] has been proposed to model the mapping from frame indices to video frames, showing competitive reconstruction performance with a very high decoding speed. When applied to video compression, the network parameters of these models are compressed through pruning, quantization, and entropy-penalization [20, 16, 18, 65] to achieve high coding efficiency. The following contributions further investigated patch-wise [5], volume-wise [43], or spatial-temporal disentangled representations [37] to improve representational flexibility. There are also methods that explicitly model the volume-wise residual [43], frame-wise residual [67], or flow-based motion compensation [66, 32, 21, 33], to enable scalable encoding and representation of longer and more diverse videos. In addition to these *index-based* approaches, other work has exploited content-specific embeddings/feature grids to provide visual prior for the network. The embeddings may be associated with single [12] or multiple resolutions [32, 30, 29, 33]. Although they hold promise in terms of low decoding complexity and competitive performance, all these aforementioned INR-based video codecs are still outperformed by state-of-the-art conventional (e.g., VVC VTM [10]) and autoencoder-based [35, 36] video codecs.

## 3 Method

Figure 2 shows the proposed NVRC framework, which follows a workflow similar to existing INR-based video compression methods, such as [13, 30, 29], but with a more advanced model compression pipeline. It trains a neural representation for a given video(s) and utilizes model compression techniques to obtain the compact representation (with compressed network parameters) of the video(s). Specifically, in NVRC, for a target video sequence $V^{gt}$ with $T$ frames, height $H$, width $W$, and $C$ channels, i.e., $V^{gt} \in \mathbb{R}^{T \times H \times W \times C}$, a neural representation $F$ parameterized by $\theta$ is trained to map coordinates to pixel intensities such as RGB colors, in a patch-wise manner. This can be represented by:

$$V_{patch} = F_{\theta}(i, j, k), \tag{1}$$

where $i, j, k$ is the patch coordinates. $V_{patch} \in \mathbb{R}^{T_{patch} \times H_{patch} \times W_{patch} \times C_{patch}}$ is the corresponding video patch, in which $0 \leq i < \frac{W}{W_{patch}}$, $0 \leq j < \frac{H}{H_{patch}}$ and $0 \leq k < \frac{T}{T_{patch}}$. This formulation (the same as in [30]) generalizes different frameworks: when $(T_{patch}, H_{patch}, W_{patch}) = (1, 1, 1)$, the neural network maps coordinates to individual pixels [52]; when $(T_{patch}, H_{patch}, W_{patch}) = (1, H, W)$, the network maps coordinates to video frames [13].

As mentioned in Section 1, existing INR-based video codecs either split the training of the INR model and model compression [13, 30], or train the model with compression techniques applied, but not entirely in an end-to-end manner [29, 33]. Unlike these works, NVRC employs more advanced

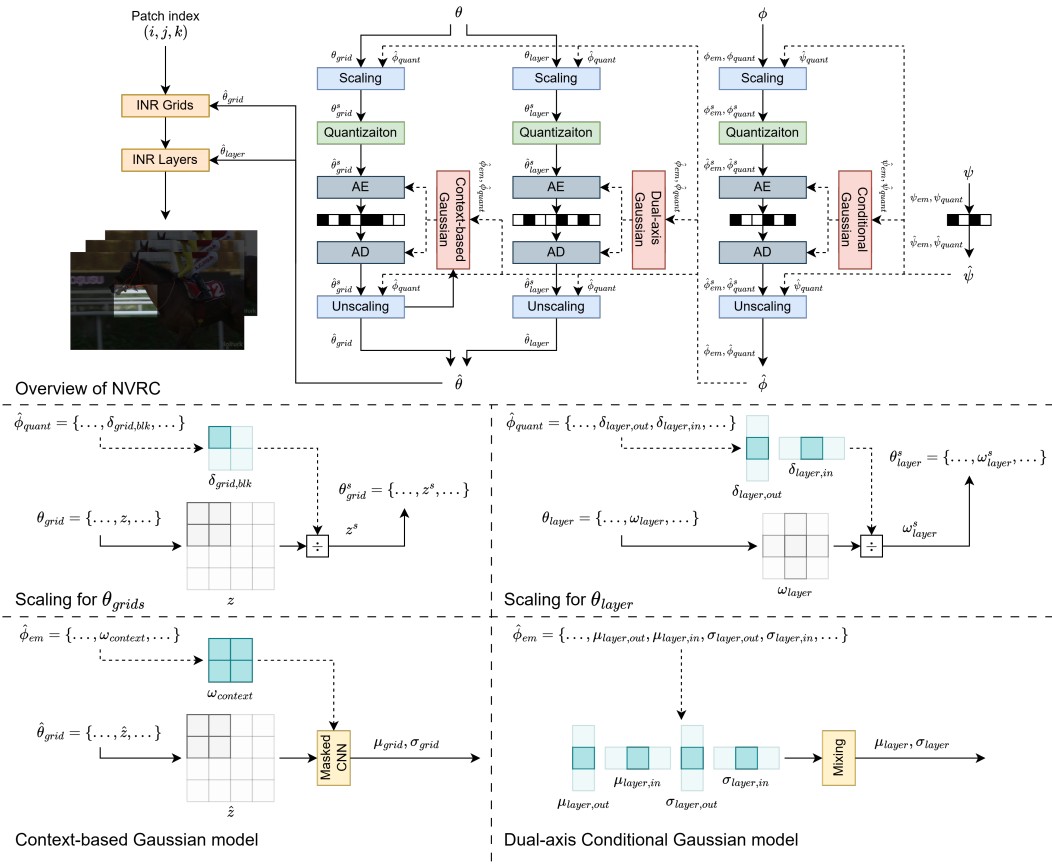

Figure 2: In NVRC, the parameters are encoded in a hierarchical structure, where (Middle-left) per-block quantization scales and (bottom-left) context-based model are utilized for encoding feature grids, and (Middle-right and bottom-right) per-axis quantization scales and dual-axis Gaussian model are applied for encoding network layer parameters.

entropy models and allows fully end-to-end optimization. Specifically, a neural representation $F$ contains a set of learnable parameters $\theta$ including feature grids parameters ($\theta_{grid}$) and network layer parameters ($\theta_{layer}$). To quantize and encode these parameters, quantization and entropy models are employed with learnable compression parameters $\phi = \{\phi_{quant}, \phi_{em}\}$. $\phi$ is determined by the distribution of the representation parameters $\theta$ in a fine-grained manner, e.g., per-group quantization scales, and can be considered as side information in compression [7]. While these parameters do improve overall coding efficiency, the introduced overhead is not negligible, particularly when the compression ratio of the representation parameters is high. Therefore $\phi$ is also quantized in this work, denoted as $\hat{\phi} = \{\hat{\phi}_{quant}, \hat{\phi}_{em}\}$, and entropy coded. Here the quantization and entropy coding are performed based on another set of learnable parameters $\psi = \{\psi_{quant}, \psi_{em}\}$, which can be simply quantized into full/half precision as $\hat{\psi} = \{\hat{\psi}_{quant}, \hat{\psi}_{em}\}$. This forms a hierarchical coding strategy for encoding these model and compression parameters $\theta$, $\phi$ and $\psi$, as illustrated in Figure 2.

All these parameters are optimized in a fully end-to-end manner based on a rate-distortion objective. Here the distortion metric $D$, e.g., mean-square-error (MSE), is calculated between a reconstructed video patch, $V_{patch} = F_{\hat{\theta}}(i, j, k)$, and the corresponding target video patch $V_{patch}^{gt}$. The rate $R$ is based on the number of bits consumed by the three levels of quantized parameters $\hat{\theta}, \hat{\phi}, \hat{\psi}$.

## 3.1 Feature grid coding

Although employing feature grids [12, 30, 29, 33, 31] for neural representations improves both convergence rate and reconstruction quality, these typically rely on a large number of parameters,

which could potentially challenge model compression techniques. To address this issue, related works utilize multi-resolution grids to improve parameter efficiency and/or perform entropy coding for the feature grid compression [12, 32, 30, 29, 33, 31].

To improve feature grid encoding, in NVRC the parameters are first partitioned into small blocks, for which different quantization parameters are applied to improve the encoding efficiency. Context-based entropy models are utilized with parallel coding

**3D block partitioning.** For a feature grid $z$ in $\theta_{grid}$, with $z \in \mathbb{R}^{T_{grid} \times H_{grid} \times W_{grid} \times C_{grid}}$, it is divided into blocks with a size of $T_{blk} \times H_{blk} \times W_{blk} \times C_{grid}$ (padding is applied if the grid size is not divisible) and produced $\frac{T_{grid}}{T_{blk}} \times \frac{H_{grid}}{H_{blk}} \times \frac{W_{grid}}{W_{blk}}$ blocks. For multi-scale grids [32, 30, 33, 29], the same block size is applied to partition the grids at each scale.

**Quantization.** With the partitioned blocks, a transformation is then applied before quantization. Here, the corresponding per-block quantization scales $\delta_{grid,blk}$ from $\hat{\phi}_{quant}$ are utilized, where $\delta_{grid,blk} \in \mathbb{R}^{\frac{T_{grid}}{T_{blk}} \times \frac{H_{grid}}{H_{blk}} \times \frac{W_{grid}}{W_{blk}} \times C_{grid}}$ is learnable, and all the scales in $\delta_{grid,blk}$ are shared by features in the same channel and in the same block in $z$. To achieve this, $\delta_{grid,blk}$ is first expanded to $\delta^{grid}$, which has the same shape as $z$, and such that:

$$\delta_{grid}[t, h, w, c] = \delta_{grid,blk}[\lfloor \frac{t}{T_{blk}} \rfloor, \lfloor \frac{h}{H_{blk}} \rfloor, \lfloor \frac{w}{W_{blk}} \rfloor, c], \tag{2}$$

where $0 \leq t < T_{grid}$, $0 \leq h < H_{grid}$, $0 \leq w < W_{grid}$ and $0 \leq c < C_{grid}$. In practice, the logarithm of $\delta_{grid,blk}$ is learned, instead of $\delta_{grid,blk}$, which ensures that the scales are non-negative.

Following [16, 43, 29, 65, 31], the scaling and quantization can then be computed by:

$$\hat{z}^s = \lfloor z^s \rceil = \lfloor \frac{z}{\delta_{grid}} \rceil, \tag{3}$$

in which $z^s$ represent the scaled parameters, $\hat{z}^s$ denote the quantized $z^s$.

The corresponding unscaling operation is defined by:

$$\hat{z} = \hat{z}^s \times \delta_{grid}, \tag{4}$$

to obtain the final quantized parameters $\hat{z}$.

**Context-based entropy model.** Although entropy coding has been used for reducing the bit rate consumed by feature grids, many implementations are only based on simple entropy models and treat the grids as ordinary network parameters [12, 32, 30, 31]. A better solution is to exploit the spatial-temporal redundancy within the feature grids, due to the inter dependent nature of the features. COOL-CHIC [33] and C3 [29] utilize context-based model and achieve efficient feature grid encoding; however, these methods are not associated with optimal rate distortion performance due to their low complexity constraints, and the use of grid entropy models for INR-based video compression has not been fully explored in the literature.

To enhance the efficiency of feature grid coding, NVRC employs context-based Gaussian models with auto-regressive style encoding and decoding processes [47] to exploit spatial-temporal redundancy within feature grids. While auto-regressive style coding is sequential, in NVRC, the feature grids at different resolutions are coded independently. Moreover, as mentioned above, each grid is partitioned into many small 3D blocks, which are also coded in parallel. Thus, the context model in NVRC has a high degree of parallelism, which enables fast coding despite of reduced amount of available context. The context-based model employs 3D masked convolution [57, 47], with parameters $\omega_{context}$ from $\hat{\phi}_{em}$, which are applied to all blocks from the same feature grid, but are not shared between grids at different scales. The context model estimates the means $\mu_{grid}$ and scales $\sigma_{grid}$ for the Gaussian distribution in a per-feature manner and uses a coder such as the arithmetic coder to code the parameters, i.e., $\hat{z}^s$. Here, the estimated means $\mu_{grid}$ and scales $\sigma_{grid}$ will also be scaled by the corresponding quantization scale $\delta_{grid}$ before applying for encoding and decoding.

### 3.2 Network layer parameter coding

Unlike the feature grids in INR models, the network parameters, such as the weights in linear and convolutional layers, are difficult to compress as there is no spatial or temporal correlation between

parameters. Existing works typically use simple entropy models for encoding these parameters [13, 30, 16, 43, 33, 65].

**2D block partitioning.** In the proposed NVRC framework, similar to feature grids, network layer parameters are also partitioned into groups prior to quantization and coding. Here we assume that the encoding of the parameters from the same input/output features/channels could benefit from sharing quantization and entropy coding parameters. For example, if an input feature/channel is zero, then the corresponding group of parameters are likely to be zeros as well. In NVRC, the quantization and entropy coding models for network parameters are designed based on this assumption, and aim to share the quantization and entropy models between parameters in the same row/column. Since there are parameters with different numbers of dimensions, e.g., 2D for linear layer weights and 4/5D for convolution weights, all layer parameters in NVRC are first reshaped into 2D tensors, where the parameters in a row correspond to the weight from the same output feature/channel, and the parameters in a column are the weights for the same input feature/channel. While existing works [13, 30] can be directly employed on partitioned parameter tensors by rows or columns, and applied per-row or per-column entropy parameters for coding, this may not be the best solution, because (i) the partitioning axis needs to be decided, (ii) the coding could benefit from sharing quantization and entropy parameters across both rows and columns. Therefore, in NVRC, the tensors are partitioned according to both axes at the same time, and the quantization and entropy parameters are learned in both axes and mixed during coding. We noticed that existing work has utilized quantization parameters on both input and output channels in different contexts [15]. Here, entropy parameters are also utilized, and both the quantization and entropy parameters are further compressed.

**Quantization.** For the weights of a layer, $\omega_{layer}$, from $\theta_{layer}$, $\omega_{layer} \in \mathbb{R}^{C_{out} \times C_{in}}$, the quantization scales $\delta_{layer} \in \mathbb{R}^{C_{out} \times C_{in}}$, is combined by two vectors of scales $\delta_{layer,out} \in \mathbb{R}^{C_{out}}$ and $\delta_{layer,in} \in \mathbb{R}^{C_{in}}$, such that:

$$\delta_{layer}[i,j] = \delta_{layer,out}[i] \times \delta_{layer,in}[j], \tag{5}$$

where $0 \le i < C_{out}$ and $0 \le j < C_{in}$. In practice, only the logarithms of $\delta_{layer,out}$ and $\delta_{layer,in}$ are stored, and quantization is performed similarly to the grid parameters (Section 3.1).

**Dual-axis conditional Gaussian model.** In NVRC, a dual-axis conditional Gaussian model is used for coding the network layer parameters. Similar to the quantization parameters mentioned above, the means $\mu_{layer}$ and the scales $\sigma_{layer}$, are represented in two per-axis parameter vectors, i.e. $\mu_{layer,out}$, $\mu_{layer,in}$, $\sigma_{layer,out}$ and $\sigma_{layer,in}$, and they are both from $\hat{\phi}_{em}$.

The combined means $\mu_{layer}$ and scales $\sigma_{layer}$ are obtained by

$$\mu_{layer}[i,j] = \mu_{layer,out}[i] \times \sigma_{layer,in}[j] + \mu_{layer,in}[j], \tag{6}$$

and

$$\sigma_{layer}[i,j] = \sigma_{layer,out}[i] \times \sigma_{layer,in}[j], \tag{7}$$

where $0 \le i < C_{out}, 0 \le j < C_{in}$. Like the quantization parameters $\delta_{layer}$, only the per-axis means $\mu_{layer,out}$, $\mu_{layer,in}$, and the logarithms of the per-axis scales $\sigma_{layer,out}$, $\sigma_{layer,in}$ are stored. Finally, the means and scales here will also (as for feature grids) be scaled by $\delta_{layer}$, before being utilized for coding $\hat{\omega}_{layer}$.

### 3.3 Coding of entropy model parameters

Since our use of more advanced quantization and entropy models will introduce additional bit rate overhead, the quantization parameters $\phi_{quant}$ and entropy model parameters $\phi_{em}$ are also quantized into $\hat{\phi}_{quant}$ and $\hat{\phi}_{em}$ and entropy coded. Here the same (as for feature grids in Section 3.1) scaling and quantization operation is applied, in which a conditional Gaussian model is used, except that the quantization scales and the means/scales for the Gaussian distribution are learned in a per-tensor manner.

### 3.4 Rate-distortion optimization

**Combined loss for NVRC.** In NVRC, the overall loss function is given below:

$$L = R + \lambda D \tag{8}$$

Here $D$ stands for the distortion calculated between the reconstructed content and the original input. $R$ is the total bitrate (bits/pixel) consumed by the quantized representation parameters $\hat{\theta}$, quantized compression parameters $\hat{\phi}$. Specifically

$$R = R_{inr} + R_{em} = \frac{1}{T \times H \times W}\left(-\sum_n^{|\hat{\theta}^s|} log_2(p_{\hat{\phi}}(\hat{\theta}^s[n])) - \sum_n^{|\hat{\phi}^s|} log_2(p_{\hat{\psi}}(\hat{\phi}^s[n]))\right) \quad (9)$$

By jointly optimizing different parameters with the combined rate-distortion loss, the trade-off between the rate and the reconstruction quality can be achieved.

**Alternating optimization.** In existing INR-based video representations and compression methods [13, 30], the distortion loss is minimized iteratively with sampling batches of frames, patches or pixels. To introduce the entropy regularization, this process has been extended [29, 65], where the rate loss is also calculated in each training step, similar to other learning-based video compression methods [40, 34]. However, in the INR-based video compression, the training is the process for over-fitting the network to a video sequence, in which the samples of each steps are from the same sequence, and the code, i.e., the INR model parameters, is the same set of parameters for all steps. Thus, it is not necessary to update the rate term in every step, especially when a significant amount of computation or memory is needed for this due to the use of entropy models. In NVRC, a more efficient training process is used, where the rate $R$ and distortion $D$ are optimized alternately. In every $K+1$ steps, the $D$ is minimized in the first $K$ steps, and where $R$ is minimized at the $K+1$-th step. Empirically, the rate loss is also scaled by $K$ to keep the rate roughly the same. Note that, the quantization step is still applied on each step, and skipping the entropy model is only possible when the quantization parameters are separated from the entropy model.

**Two-stage training.** Similar to some existing works [13, 30, 29], NVRC is also trained in two stages.

In Stage 1, to optimize $L$, the non-differentiable quantization operation needs to be emulated through a differentiable approximation during training. Recent work [29] has shown that a soft-rounding operation with an additive Kumaraswamy noise can be used to replace quantization for neural representation training. While in [29], this is applied only to feature grids, we extend this idea and apply it to both feature grids and network parameters in the first stage of training. Compared to [29], soft-rounding with higher temperature (0.5 to 0.3) is used in NVRC, as the original, low temperature (e.g. 0.3 to 0.1 in [29]) for both feature grids and network parameters will lead to training difficulty due to the large variance of the gradients.

In the second stage, instead of using soft-rounding, following [30, 31], Quant-Noise [53] is used to fine-tune the neural representation, as we empirically found that Quant-Noise remains stable with different hyper-parameter settings and is suitable for high quantization levels.

## 4 Experiment

### 4.1 Experiment Configuration

**Evaluation database.** To evaluate the performance of the proposed NVRC framework, we conducted experiments on the UVG [45] and MCL-JCV [59] dataset. The UVG dataset includes 7 video sequences with 300/600 frames, while the MCL-JCV dataset consists of 30 video clips with 120-150 frames. All sequences are compressed at their original resolution in this experiment. We also provide the result of JVET-CTC dataset Class B [9] in the *Appendix*.

**Implementation details.** NVRC is a new framework focusing on INR model compression, which can be integrated with any typical INR-based models. To test its effectiveness, we employed one of the latest INR network architectures, HiNeRV [30], and integrated it into our NVRC framework. This INR model has been reported to provide competitive performance compared to many standard and end-to-end codecs for the video compression task. Minor adjustments have been made on top of HiNeRV in terms of the network structure and the training configuration (see *Appendix* for details) - rate points are now obtained by both turning the scale of the neural representation and the $\lambda$ value. The model is trained for 360 or 720 epochs in the first stage and 30 or 60 epochs in the second stage, depending on the UVG [45] and MCL-JCV [59] datasets, due to the differing lengths of the sequences.

Table 1: BD-rate results on the UVG dataset [45].

| Color Space | Metric | x265 (*veryslow*) | HM (*RA*) | VTM (*RA*) | DCVC-HEM | DCVC-DC | HiNeRV | C3 | HNeRV-Boost |
|---|---|---|---|---|---|---|---|---|---|
| RGB 4:4:4 | PSNR | -73.74% | -50.38% | -23.42% | -40.57% | -31.20% | -50.16% | -66.86% | -66.45% |
| | MS-SSIM | -80.65% | -67.38% | -49.75% | -6.97% | -11.75% | -44.27% | -76.59% | -78.01% |
| YUV 4:2:0 | PSNR | -66.89% | -42.50% | -12.96% | - | -33.98% | - | - | - |
| | MS-SSIM | -59.38% | -38.20% | -15.04% | - | -39.12% | - | - | - |

Table 2: BD-rate results on the MCL-JCV dataset [59].

| Color Space | Metric | x265 (*veryslow*) | HM (*RA*) | VTM (*RA*) | DCVC-HEM | DCVC-DC | HiNeRV | C3 | HNeRV-Boost |
|---|---|---|---|---|---|---|---|---|---|
| RGB 4:4:4 | PSNR | -51.61% | -13.88% | 36.91% | -2.97% | 13.40% | -31.69% | -42.23% | -59.80% |
| | MS-SSIM | -66.83% | -41.01% | -6.39% | -21.64% | 33.07% | -41.62% | -49.23% | -83.36% |
| YUV 4:2:0 | PSNR | -49.02% | -13.19% | 40.80% | - | 3.75% | - | - | - |
| | MS-SSIM | -43.00% | -12.61% | 33.09% | - | -9.89% | - | - | - |

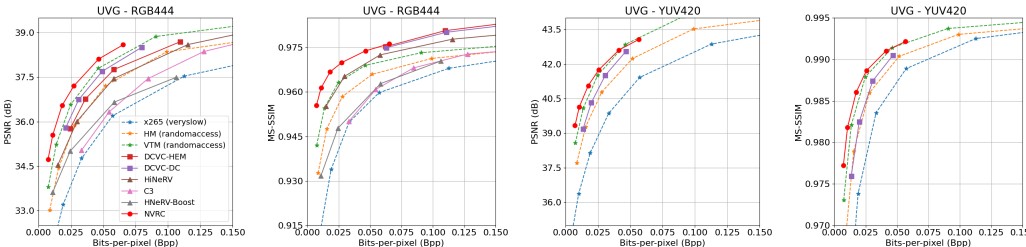

Figure 3: Average rate quality curves of various tested codecs on the UVG dataset [45].

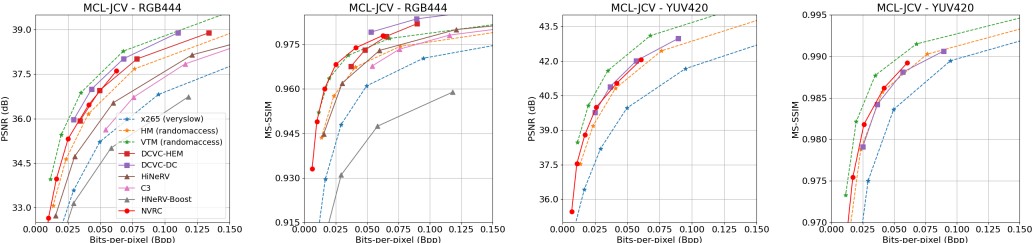

Figure 4: Average rate quality curves of various tested codecs on the MCL-JCV dataset [59].

**Benchmark methods.** Conventional codecs, x265 [1] with the *veryslow* configuration, HM-18.0 [50] and VTM-20.0 [11] with the Random Access configuration, are used for benchmarking, together with two recent learned video codecs, DCVC-HEM [35], DCVC-DC [36]. Three state-of-the-art INR-based codecs, including the original HiNeRV [30], C3 [29] and HNeRV-Boost [65] have also been included in this experiment. All results are produced by the open source implementations.

**Evaluation methods.** The evaluation was performed in the RGB color space (for comparing both conventional codecs and learning-based methods) with the BT.601 color conversion, and in the original YUV420 color space (for comparing both conventional methods and the learning-based methods that support this feature in their public implementations). PSNR (RGB/YUV 6:1:1) and MS-SSIM (RGB/Y) are used here to assess video quality, based on which Bjøntegaard Delta Rate figures are calculated against each benchmark codec.

## 4.2 Results and discussion

Figure 3-4 and Table 1-2 provide the results for the proposed NVRC model and the benchmark methods. It can be observed that when tested in the RGB 4:4:4 color space (as in many learning-based works), NVRC significantly outperforms the original HiNeRV model [30], with an average coding gain of 50.16%, measured by PSNR. Similar improvement has also been achieved against other INR-based methods including HNeRV-Boost [65] and C3 [29]. Moreover, NVRC also offers better performance compared to latest MPEG standard codec VVC VTM (Random-Access) [11] on the UVG dataset [45], with a 23.4% average coding gain based on PSNR. To the best of our knowledge, it is the first INR-based video codec outperforming VTM. Compared to state-of-the-art learned video coding methods, NVRC also exhibits superior or comparable performance to DCVC-HEM [35] and

Table 3: Complexity results of NVRC with the UVG dataset [45]. Encoding and decoding FPS are measured by the number of training steps/evaluation steps per second performed by the INR model with 1080p inputs/outputs. The model compression MACs and encoding/decoding time are measured by the steps for performing quantization and entropy coding. The complexity figures are calculated based on NVIDIA RTX 4090 GPU with FP16.

| Rate point | Frame MACs/Enc FPS/Dec FPS | Model compression MACs/Enc time/Dec time |
|---|---|---|
| 1-2 | 359.6G/6.4/21.0 | 25.2G/22.9s/37.0s |
| 3-4 | 842.8G/3.6/15.1 | 50.4G/29.6s/44.8s |
| 5-6 | 1929.0G/2.2/9.7 | 100.8G/43.4s/53.7s |

Table 4: Ablation studies on the UVG dataset [45]. Results are BD-rates.

| Metric | NVRC | (V1) | (V2) | (V3) | (V4) | (V5) |
|---|---|---|---|---|---|---|
| PSNR | 0.00% | 13.04% | 11.06% | 23.37% | 30.84% | 14.42% |
| MS-SSIM | 0.00% | 13.84% | 10.24% | 23.88% | 30.07% | 8.54% |

DCVC-DC [36] on the UVG [45] and MCL-JCV [59] datasets, respectively.. When evaluated in the YUV 4:2:0 color space, NVRC still offers superior performance as for RGB 4:4:4 color space, outperforming most benchmarked methods based on PSNR and MS-SSIM. It should be also noted that INR-based video codecs do not require offline training on large-scale datasets, whereas other learning-based methods do. Qualitative results are provided in Figure 1 in terms of visual comparison between the content reconstructed by NVRC and HiNeRV.

## 4.3 Computational complexity

The complexity figures of NVRC with the UVG dataset [45] are provided in Table 3. When compared to the original HiNeRV [30], the proposed method (with HiNeRV as its INR network) is associated with increased computational complexity. However, the MACs figure is still significantly lower than that of other learning-based video codecs (e.g., DCVC-DC [36]), which allows faster decoding. It should be noted that the complexity figures shown here are obtained based on research source code that has not been optimized for latency. The actual latency of INR and entropy coding can be further reduced by (1) optimizing the implementation of the INR and entropy models, (2) performing lower precision computation, and (3) implementing parallel decoding between different resolution feature grids.

## 4.4 Ablation study

To evaluate the contribution of the main components in NVRC, an ablation study was performed based on the UVG dataset [45], using the configurations in Section 4.1, but four rate points for each variant.

**Alternative entropy model settings.** We compared different combinations of entropy models for encoding feature grids $\theta_{grid}$ and network parameters $\theta_{layer}$: Context model + dual-axis conditional Gaussian model (Default setting in NRVC), (V1) Context model + per-tensor conditional Gaussian model, (V2) per-tensor conditional Gaussian model + dual-axis conditional Gaussian model, (V3) per-tensor conditional Gaussian model + per-tensor conditional Gaussian model.

**Hierarchical parameters coding.** In NVRC, the quantization parameters $\phi_{quant}$ and the entropy model parameters $\phi_{em}$ are also entropy coded. To verify this, we created another variant (V4) with $\phi_{quant}$ and $\phi_{em}$ not coded but stored in half-precision.

**Learned quantization steps.** We also compared the use of learned quantization steps $\phi_{quant}$ and $\psi_{quant}$ (Default setting in NRVC) and (V5), a new variant with fixed quantization steps for grids, where the log-step size is set to $-4$.

Table 4 shows the ablation study results, in terms of the BD-rate values against the original NVRC. These figures confirmed the contribution of the tested components in the NVRC framework.

In addition, we conducted experiments to evaluate the effects of fully end-to-end optimization and alternating optimization on selected challenging sequences from the UVG dataset (Jockey and ReadySetGo) [45]. When removing fully end-to-end optimization, the variant without rate loss in the first stage exhibits up to a 35% BD-rate increase compared to the proposed model. However, this loss diminishes if the number of epochs in the second stage increases. With the proposed alternating optimization, we did not observe any noticeable difference in performance. Nevertheless, without alternating optimization, the training step time can increase by up to 40% under our experimental settings.

## 5 Conclusion

In this paper, we present NVRC, a new INR-based video compression framework with a focus on representation compression. By employing novel entropy coding and quantization models, NVRC significantly improved coding efficiency and allows real end-to-end optimization for the INR model. The experimental results show that NVRC outperforms all the benchmarked conventional and learning-based video codecs, in particular with a 23% bitrate saving against VVC VTM (Random Access) [11] on the UVG database [45]. This is the first time an INR-based video codec has obtained this achievement.

## Acknowledgments and Disclosure of Funding

This work was supported by UK EPSRC (iCASE Awards), BT, the UKRI MyWorld Strength in Places Programme and the University of Bristol. Part of the computational work was also supported by the facilities provided by the Advanced Computing Research Centre at the University of Bristol.

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

# A  Appendix

## A.1  Additional experiments

Table 5: BD-rate results on the JVET-CTC Class B dataset [9].

| Color Space | Metric | x265 (*veryslow*) | HM (*RA*) | VTM (*RA*) | DCVC-DC |
|---|---|---|---|---|---|
| RGB 4:4:4 | PSNR | -68.70% | -35.53% | 7.48% | -13.85% |
| | MS-SSIM | -84.69% | -65.51% | -42.60% | -18.81% |
| YUV 4:2:0 | PSNR | -66.33% | -34.49% | 9.57% | -24.25% |
| | MS-SSIM | -66.02% | -39.99% | -6.79% | -36.94% |

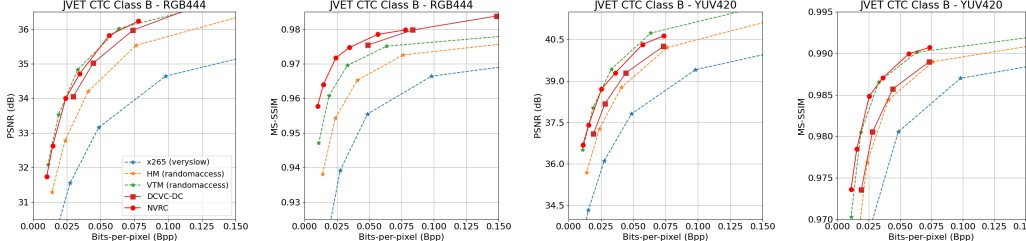

Figure 5: Average rate quality curves of various tested codecs on the JVET-CTC Class B datasets [9].

In addition to the main paper experiments, here we provide the results of NVRC with the JVET-CTC test sequences for VTM [9]. Similar experiments configurations as in the main paper have been used, where x265 [1] with the configuration, HM-18.0 [50] and VTM-20.0 [11] with the Random Access configuration, and DCVC-DC [36] have been used as the baseline models. Similar to the primary experiment results, our proposed NVRC consistently outperforms the baseline models in most cases.

## A.2  NVRC configurations at different scales.

**Neural representation.**  In NVRC, the INR architecture is based on HiNeRV [30], with minor modifications. The differences include: (1) due to the enhanced capability of feature grids coding, feature grids in NVRC are larger for better capability on capturing dynamic contents. (2) The bilinear interpolation of the input encoding is performed after the stem convolution layer, while in the original HiNeRV it was before applying convolution. This improves the performance marginally. (3.) The hyper-parameter selection between different scales has been simplified, where only the number of channels of the network layers and the feature grids change with scales.

The hyper-parameters of the neural representation in NVRC are provided in Table 7 and 8. It is noted that in NVRC, only the input feature grids in HiNeRV are coded by the context model, while the local grids are simply considered as general layer parameters, as they only account for a small number of parameters.

Table 6: Comparison between NVRC to existing works with entropy regularization.

| Method | Stage 1 | Stage 2 | Grid EM | Layer EM | Quant/EM parameter sharing | Multi-level coding |
|---|---|---|---|---|---|---|
| Zhang et al. [66] | D | R+D | N/A | Uniform | per-channel | No |
| Gomes et al. [16] | D | R+D | N/A | Neural Network | per-weight | No |
| Maiya et al. [43] | R+D | R+D | N/A | Neural Network | per-weight | No |
| Kim et al. [29] | R+D | R+D | Context | Laplace* | No | No |
| Leguay et al. [33] | R+D | - | Context | Laplace* | No | No |
| Zhang et al. [65] | D | R+D | Gaussian | Gaussian | per-weight | No |
| Ours | R+D | R+D | Context | Gaussian | per-block/per-axis | Yes |

*: Applied only after training.

**Context-based entropy model.**  The context-based entropy model in NVRC has an autoregressive style coding process. It is based on masked convolution [57, 47]. It contains 3 blocks, where each block contains a Layer Normalization layer [4], 3D convolution and GeLU activation [24] (except

Table 7: NVRC configurations.

| | S1 | S2 | S3 | S4 |
|---|---|---|---|---|
| Number of parameters | 2.14M | 6.35M | 14.19M | 31.41M |
| Channels | (224, 112, 56, 28) | (336, 168, 84, 42) | (512, 256, 128, 64) | (768, 384, 192, 96) |
| kernel size | | | $3 \times 3$ | |
| Expansion ratios | | | (4, 4, 4, 4) | |
| Depths | | | (3, 3, 3, 1) | |
| Strides | | | (3, 2, 2, 2) | |
| Stem kernel size | | | $3 \times 3 \times 3$ | |
| Grid sizes | $T_{grid} \times 45 \times 80 \times 1$ | $T_{grid} \times 45 \times 80 \times 2$ | $T_{grid} \times 45 \times 80 \times 4$ | $T_{grid} \times 45 \times 80 \times 8$ |
| Grid levels | | | (4) | |
| Grid scaling ratios | | | (2, 2, 2, 0.5) | |
| Local grid sizes | $T \times 4$ | $T \times 8$ | $T \times 16$ | $T \times 32$ |
| Local grid levels | | | (3) | |
| Local grid scaling ratios | | | (2, 0.5) | |

$T$: the number of video frames
$T_{grid}$: 200 for UVG [45]/JVET-CTC Class B [9], 50 for MCL-JCV [59]

for the final output). The context-based model is independent between channels, and has been implemented as with depth-wise convolution. The kernel size is 5 and the width is 8. The output of the context-based model is the means and log scales of the Gaussian distribution.

**Feature grids/layer parameters partitioning.** In NVRC, the feature grid parameters are partitioned into blocks, and the quantization parameters are shared within each block, while the layer parameters are partitioned into rows and columns, with both of the quantization and entropy parameters shared. For feature grid parameters, the block size used is $16 \times 8 \times 8$, which is a relative small size but can provide high degree of parallelism for the auto-regressive coding process. For the layer parameters, there are different size of parameters in the neural representation, where some of them are very small and could be too costly to include the per-column/row parameters. Thus, for those light weight parameters, either single-axis or per-tensor quantization/entropy parameters are used. In particular, we use the per-column/row quantization and entropy parameters if the number of parameters on the column/row is at least 128.

### A.3 Experiment configurations

In the experiments, the configurations of NVRC are based on [30]. The training is performed by sampling patches from the target video, where the patch size is $120 \times 120$, and the batch size of each training step is 144 patches (equal to 1 frame). For learning RGB output, the distortion loss is:

$$D = 0.7 \times \text{L1}_{RGB} + 0.3 \times (1 - \text{MS-SSIM}_{RGB}) \qquad (10)$$

where the MS-SSIM has a reduced window size ($5 \times 5$) due to the training in small patches. For YUV output, NVRC is trained the YUV444 setting, to avoid changing the model architecture, but evaluation is in YUV420. In related works, different loss functions for YUV outputs have not been studied. Here we use the loss

$$D = 0.99 \times (\text{MSE}_Y^{6/8} \times \text{MSE}_U^{1/8} \times \text{MSE}_V^{1/8}) + 0.01 \times (1 - \text{MS-SSIM}_Y), \qquad (11)$$

which align with both the commonly used PSNR-YUV (6:1:1) and MS-SSIM (Y) metrics. While we did not thoroughly study the weighting between two terms, we found that this ratio offers both a good PSNR and MS-SSIM performance.

The learning rates in Stage 1 and Stage 2 are 2e-3 (or 1e-3 for rare case which the training is less stable) and 1e-4, where the cosine decay is applied for scaling the learning rate with a minimum learning rate of 1e-4 and 1e-5, respectively. The norm clipping with 1.0 is applied. L2 regularization of 1e-6 is applied to improve the numerical stability, as we observe that the norm of weight could grow too large in some cases. The magnitude of L2 regularization linearly decays in the first stage and is not applied in the second stage, to avoid under-fitting. For the soft-rounding and the Kumaraswamy noise [29] in Stage 1, the temperatures and the noise scale ratio scale from 0.5 to 0.3 and 2.0 to 1.0, respectively. For Quant-Noise [53] in Stage 2, the noise ratio scales from 0.5 to 1.0. Note that we follow [30], where rounding, instead of the straight-through estimator (STE) [8], is applied to the quantized variables. $R$ is optimized once for every 8 steps of $D$.

For the conventional codecs, we used a QP range of 16–44 for x265 [1] and 16–36 for HM [50] and VTM [11], with QP values selected at intervals of 4. The results for HiNeRV [30] and HNeRV-Boost [65] are obtained by training with their provided implementations, with the original configurations.

Table 8: NVRC representation scales and regularization ($\lambda$) configurations for the UVG [45], MCL-JCV [59] and JVET-CTC Class B [9] datasets (RGB/YUV).

| Rate point | UVG | | MCL-JCV | | JVET-CTC Class B | |
|---|---|---|---|---|---|---|
| | RGB | YUV | RGB | YUV | RGB | YUV |
| 1 | S2, $\lambda = 1.0$ | S2, $\lambda = 32.0$ | S1, $\lambda = 0.25$ | S1, $\lambda = 8.0$ | S2, $\lambda = 1.0$ | S2, $\lambda = 32.0$ |
| 2 | S2, $\lambda = 2.0$ | S2, $\lambda = 64.0$ | S1, $\lambda = 0.5$ | S1, $\lambda = 16.0$ | S2, $\lambda = 2.0$ | S2, $\lambda = 64.0$ |
| 3 | S3, $\lambda = 4.0$ | S3, $\lambda = 128.0$ | S2, $\lambda = 1.0$ | S2, $\lambda = 32.0$ | S3, $\lambda = 4.0$ | S3, $\lambda = 128.0$ |
| 4 | S3, $\lambda = 8.0$ | S3, $\lambda = 256.0$ | S2, $\lambda = 2.0$ | S2, $\lambda = 64.0$ | S3, $\lambda = 8.0$ | S3, $\lambda = 256.0$ |
| 5 | S4, $\lambda = 16.0$ | S4, $\lambda = 512.0$ | S3, $\lambda = 4.0$ | S3, $\lambda = 128.0$ | S4, $\lambda = 16.0$ | S4, $\lambda = 512.0$ |
| 6 | S4, $\lambda = 32.0$ | S4, $\lambda = 1024.0$ | S3, $\lambda = 8.0$ | S3, $\lambda = 256.0$ | S4, $\lambda = 32.0$ | S4, $\lambda = 1024.0$ |

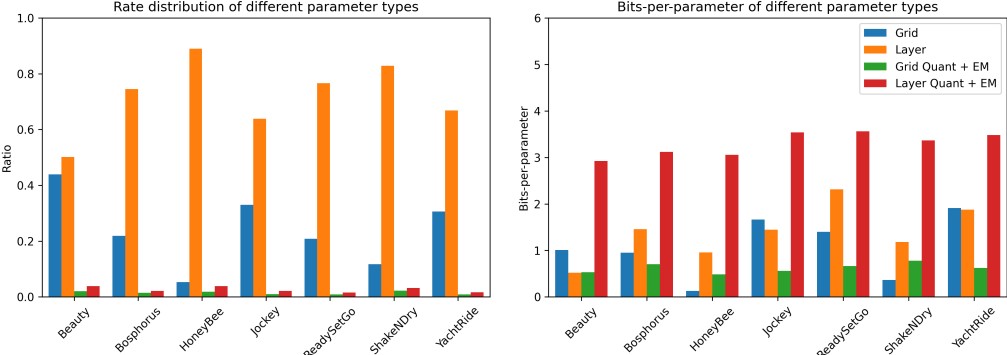

Figure 6: (Left) Rate distribution and (Right) bits-per-parameter of different parameter types.

### A.3.1 Rate distribution

In Fig 6 (left), we provide the rate distribution of different types of parameters (feature grids, network layer parameters, quantization and entropy coding parameters for feature grids/layer parameters). The data is collected from the lowest rate point models with the UVG dataset [45]. It can be observed that, in general, the quantization and entropy coding parameters contribute to a very small amount of the total bitrate, where the ratio between the feature grids and the layer parameters vary between different video sequences. In Figure 6 (right), we further provide the bits-per-parameter data for different video sequences. In this very low bitrate rate point, NVRC is capable of learning parameter distributions with very low bits-per-parameters, and it also varies between sequences. For example, on some sequences with larger motion (Jockey and YachtRide), the bits-per-parameters of the feature grids can be doubled, but for some relatively static sequences, the bits-per-parameters of the feature grids is nearly zero.

### A.4 Positive Impacts

NVRC is the first INR-based codec which outperforms VVC VTM [11] with a 23% coding gain on the UVG database [45]. It will potentially contribute to the next generation of video coding standards, and improve current video streaming services if deployed in practice.

### A.5 Limitations

**Encoder complexity.** As the implementation of NVRC is based on sophisticated neural networks, it requires substantial computational resources, in particular at the encoder. This is a common issue with many INR-based approaches that require content overfitting during encoding. It makes this type of approaches unsuitable for real-time encoding scenarios like video conferencing. This also results in increased energy consumption and a negative environmental impact. Future work should focus on reducing the complexity of this model.

**Latency.** As INR-based video codecs require overseeing all frames of a video sequence at the same time during encoding, the system latency become more longer compared to conventional and some end-to-end learned video codecs which perform per-frame encoding. This prevents these INR models from adoption in practical application scenarios.

**Reproducibility.** In this paper, we have not studied reproducibility, which is a critical issue for the practical application of deep video compression with entropy coding. While our experiments focus on floating-point operations, the operations in the proposed method can also be implemented, for example, using integer operations [6], which can ensure reproducibility.

