# OpenReview forum: "NVRC: Neural Video Representation Compression"
_NeurIPS.cc/2024/Conference — NeurIPS 2024 poster_

### Official Review · Reviewer_m68p · 2024-06-23

**Soundness:** 3
**Presentation:** 3
**Contribution:** 3
**Rating:** 6
**Confidence:** 4

**Summary:**

This paper proposes a INR-based video compression framework, Neural Video Representation Compression (NVRC), targeting compression of the implicit representation.
Based on the proposed novel entropy coding and quantization models, NVRC is able to optimize an INR-based codec in a fully end-to-end manner.
They also proposed a model compression framework for coding all the network, quantization and entropy model parameters hierarchically.
Experiments show that NVRC outperforms many conventional and learning-based codecs, with a 24% overall coding gain over VTM (Random Access) on the UVG dataset, measured in PSNR.

**Strengths:**

The manuscript is well written and easy to follow.
Though the components exist in previous learned compression works, the proposed overall framework has novelty, which builds an elegant and efficient dependency among grid parameter, neural network parameter and hyperprior.
The rate distortion performance is outstanding, making INR based methods better than VTM.

**Weaknesses:**

Only UVG dataset is evaluated in the experiment. Please consider using more dataset as previous works do.

In VAE based deep video compression, there exists a line of works like ‘Bit allocation using optimization, ICML 2023’, which is based on overfitting the latent representation to individual input video/GoP during encoding. This line of research is closely related to INR based video compression and should be discussed in the related works part.

The complexity in Table 2 should be compared with previous methods like figure 3.

The components used in this paper exist in previous works and should be discussed as previous works, including group based quantization (Fully quantized network for object detection, CVPR 2019), patch or block based parallel context model (Riddle: Lidar data compression with range image deep delta encoding, CVPR 2022; Checkerboard context model for efficient learned image compression, CVPR 2021).  It is better to highlight the technical novelty in each component by comparing with those previous methods.

Minor:
L95, [11] is cited twice.

**Questions:**

Please refer to the weakness part. Especially about the complexity and technical novelty comparision with previous works.

**Limitations:**

The limitation is addressed in the manuscript.

---

> ### Author Rebuttal · Authors · 2024-08-07
>
> ***Q: Evaluation on more datasets***
>
> A: Thank you for this suggestion. We agree that evaluation on more datasets is important, and we have included additional results (MCL-JCV and JVET CTC Class B, RGB setting) in the rebuttal results (see the attached pdf). Here we used the JVET CTC Class B rather then JCT-VC CTC Class B (HEVC-B) as the former is the latest HD testset for VVC. It can be observed that promising results have been observed in both dataset. The performance of NVRC is significantly better than HiNeRV, the previous best INR-based method in the MCL-JCV dataset. It is noted that the performance of NVRC with the MCL-JCV dataset is worse compare with the other datasets. We believe that this is mainly due to the shorter length of the sequences, and the above observation is consistent with previous work [24]. Full results on these datasets will also be provided in the final version of the paper.
>
> ***Q: Discussing related works on VAE based deep video compression***
>
> A: We appreciate this suggestion. In the paper, we have referred to related work on overfitting-based VAE for image [55] and video [52] compression, but we agree that it would be beneficial to include further contributions in this class including the paper suggested by the reviewer [Xu2023].
>
> ***Q: Complexity versus Rate plots ***
>
> A: We agree that complexity versue rate plots will provide a better illustration of performance. We have included these in the rebuttal results (see the attached pdf) and will of course include them in the revised paper.
>
> ***Q: Technical novelty in each component over existing works.***
>
> A: Thank you for highlighting this. We agree that group-based quantization and patch/block-based context models have been employed in previously published papers, and we will include discussion of this in the revised paper.
>
> Regarding the novelty of our approaches over these works, although the aforementioned components do exist in the literature, existing INR-base video codecs have rarely employed these models, especially to combine learned quantization with different entropy models.
>
> In terms of the use of quantization and entropy models, we would like to highlight that there is a gap in the design of these components between conventional end-to-end approaches and INR-based compression models; existing INR-based models only opt for simple solutions, e.g., conditional Gaussian or simple auto-regressive models. Our work exploits the use of more complex models, i.e., the axis-based conditional Gaussian and block-based auto-regressive models, which offer advantages over existing INR-based approaches. However, we do understand that these entropy models are not SOTA choices when considering all learning-based compression models, e.g. the checkerboard model [He2021].
>
> While our technical novelties include the use of both better quantization and entropy models for INR-based video compression, more importantly, we have combined different quantization and entropy models into a fully optimized framework and propose encoding the whole model (the INR + quantization/entropy models) in a hierarchical manner. Previous work has usually utilized a single entropy model for encoding different types of parameters. For example, [23, 27] only use a learned context model for encoding latent grids but employs non-learned quantization and entropy model for coding network parameters; [57] only uses a conditional Gaussian model for encoding all types of parameters. In our work, we use group-based quantization for network parameters and block-based context models for feature grid encoding. This offers better coding efficiency, as the parameters are encoded by a class of models that are likely to be more efficient. Furthermore, in INR-based compression, introducing more complex quantization and compression models can introduce significant overhead, because the parameters of these models have also to be transmitted to the decoder side. Thus, we introduce the hierarchical parameter coding structure into INR-based compression, which enables the use of more complicated quantization and compression models.
>
> We will include the above discussion in the paper.
>
> ***Q: [11] is cited twice***
>
> A: Thank you for pointing out this issue. We will fix this in the camera-ready version.
>
> ***Refs***
>
> [Xu2023] Xu, Tongda, et al. "Bit allocation using optimization." International Conference on Machine Learning. PMLR, 2023.
>
> [He2021] He, Dailan, et al. "Checkerboard context model for efficient learned image compression." Proceedings of the IEEE/CVF Conference on Computer Vision and Pattern Recognition. 2021.

---

> > ### Comment · Reviewer_m68p · 2024-08-09
> >
> > Thanks for the relpy, I increase my score from ba to wa after considering all the reviewer comments.

---

### Official Review · Reviewer_BGkd · 2024-07-08

**Soundness:** 1
**Presentation:** 2
**Contribution:** 2
**Rating:** 5
**Confidence:** 5

**Summary:**

This paper focuses on the implicit neural representation for video compression, where several modifications are applied for the coding of feature grid and network layer. An enhanced training pipeline is also applied.

**Strengths:**

1. The claimed performance over traditional codec and VAE-based codec is impressive.

**Weaknesses:**

1. This paper claims the first fully end-to-end optimized INR-based framework. However, the problem of joint rate-distortion optimization has been addressed for INR-based compression in several existing works, including [13] and [Ref1] (missing in this paper). As these two papers have already been dedicated to enabling joint rate-distortion optimization in INR-based image compression, it is important to tell the difference in this paper when compared with these two previous works. I know these two previous works are for image, while this paper focuses on video. If the author think the difference between image and video in this part is large, the author should highlight the challenge and the corresponding solution of joint rate-distortion optimization for video when compared with image.

[Ref1] Compression with Bayesian Implicit Neural Representations. NeurIPS 2023.

2. In Fig. 3 UVG-YUV 420, the PSNR of baseline DCVC-DC is about 40 dB at 0.03 bpp. However, in the Fig. 7 of DCVC-FM paper [https://github.com/microsoft/DCVC/tree/main/DCVC-FM], the PSNR of DCVC-DC is obviously larger than 41 dB at 0.03 bpp. So, for the same model, what is the reason for this large RD difference?

3. The results on MCL-JCV and HEVC B are missing. The ablation study is only based on two videos, which is not reliable.

Currently I think this paper has very good performance. If the author can address my concerns and questions, I will consider increasing my rating.

**Questions:**

1.  In Table 1, what is the anchor method? Where is the result of the proposed NVRC in Table 1?

2.  In Table 2, the Frame Enc FPS can be as high as 6.4 FPS, so can the NVRC encode a 600-frame UVG video within 100 seconds? From my understanding, the INR-based solution is quite slow for encoding. The author should give the encoding time from RGB video to binary bit-stream, and the decoding time from binary bit-stream to RGB video, where the actual arithmetic coding should be performed, especially when considering NVRC has auto-regressive model.

**Limitations:**

no negative societal impact

---

> ### Author Rebuttal · Authors · 2024-08-07
>
> ***Q: The difference between NVRC and [13]/[Guo2023]****
>
> A: We thank the review for highlighting this point. We agree that there are multiple INR-based codecs that focus on joint rate-distortion optimization, and will describe these in the revised paper.
>
> When we refer to a "fully end-to-end optimized framework" in the paper, we imply that i) the approaches should be optimized using a rate-distortion loss during the whole encoding process, and ii) all components with parameters that will be entropy coded, and their related quantization/entropy parameters should be optimized with according to the rate-distortion objective. Based on this definition, we believe that [13] is not fully optimized. Specifically, in [13], the rate term only contributes to the second stage optimization. In contrast, our approach is fully end-to-end optimized, because:
>
> 1) The RD loss is used throughout the encoding process, including both Stage 1 and 2. This is important because i) it provides improved performance (as shown in the additional ablation study in the rebuttal, V6); ii) it could also provide a more streamlined solution, as pointed out in a recent work [23], when incorporating rate distortion optimization in the first stage of training, the second stage is not mandatory as it only contributes to very little gain (we have observed a similar characteristic in our experiment).
>
> 2) The total rate is also jointly optimized in our work, which includes all the parameters that are entropy coded and have non-negligible overhead. In contrast, previous works [23, 27, Leguay2023] only include some of them. In [Leguay2023], only the latent grids are optimized with RD optimization, while the network parameters are entropy coded without optimizing the rates. In their limitation section (Sect V.), they pointed out that the network parameters actually cost a high overhead (up to >30\% bit-rate) especially at low rates.
>
> In [Guo2023], the focus was solely on image compression. We also observed that the framework in [Guo2023] differs significantly from those used in INR-based video compression [10, 13, 17, 23, 24, 26, 27]. While the former encodes the parameters without quantization, the latter performs quantization followed by entropy coding. Our approach follows the second method, which is mainly adopted in INR-based video coding tasks.
>
> Regarding the difference between INR-based image and video compression: for video compression, INR-based approaches usually utilize a larger network, due to both the larger size of the signal instance and the higher compression ratio required. For example, NeRV-based models [10, 13, 17, 23, 24, 26, 27] typically contain millions of parameters. Due to the larger size of the models, more complicated compression techniques, such as fine-grained quantization and entropy parameters, are beneficial. Therefore, INR-based image and video compression tasks introduce different challenges, and we think that the fully end-to-end optimization could play a more important role for the video compression task compared to image coding, which allows us to achieve improved overall rate distortion performance.
>
> ***Q: DCVC-DC results in Fig. 3***
>
> A: We appreciate the reviewer's attention to detail. We have identified that this inconsistency is due to the incorrect use of the model checkpoints. We have corrected the results (as included in the rebuttal results) and will update them in the revised paper. We have also cross-checked the remaining results and confirmed their correctness.
>
> ***Q: Evalution (and ablation study) on more content***
>
> A: This is a valid observation. We have provided preliminary coding results on the MCL-JCV and JVET CTC Class B datasets in the rebuttal (see the attached pdf) and will include full results in the revised paper. Here we used the JVET CTC Class B rather then JCT-VC CTC Class B (HEVC-B) as the former is the latest HD testset for VVC. We agree that performing the ablation study on more sequences is important. However, due to the limited time available, we have not been able to generate these additional ablation study results during the rebuttal. We commit to providing full results on the whole UVG dataset in the paper.
>
> ***Q: Anchor methods in Table 1***
>
> A: In Table 1, we compare NVRC (test) against each of the baseline models (anchor) to calculate the BD-rate results. We will make this clear in the paper.
>
> ***Q: Encoding time in Table 2***
>
> A: In Table 2, we provide the encoding (training) speed in terms of frames per second (FPS) per each training epoch. Therefore, the total enc time = (#frames x #epoches)/enc FPS, e.g., the total time for encoding a 600 frame sequences with 390 epochs is around 10 hours, when the encoding speed is 6.4 FPS.
>
> We reported the enc speed in this manner instead of the total time, for two reasons: i) this is commonly used in related works (e.g. [23] reported enc time per 1K steps and [24] reported the time per step), ii) the number of epochs of INR-based approaches is different with varying quality.
>
> We have reported both the i) entropy en/decoding times for the whole model ('Model Compression Enc/Dec Time' in Table 2), ii) INR decoding time for reconstructing a video frame ('Frame Dec Time' in Table 2). For example, it takes 37 seconds for entropy decoding the smallest model, and this decoding process only needs to be performed once, and the decoded model can be used for reconstructing up to 600 frames in our experiment. Additional parallelism can be further achieved for faster coding with the autoregressive model, but we have not implemented it in our experiment: i) parallel en/decoding can be enabled because different resolution grids are independently coded, ii) parallel entropy decoding and frame reconstruction is also possible, as the decoding a single video frame does not require the full grids to be decoded.
>
> We will clarify these points in the final paper and report the total en/decoding times as suggested by the reviewer.

---

> > ### Comment · Reviewer_BGkd · 2024-08-12
> > **Response**
> >
> > I increased my rating.

---

### Official Review · Reviewer_YLxP · 2024-07-12

**Soundness:** 3
**Presentation:** 3
**Contribution:** 3
**Rating:** 7
**Confidence:** 4

**Summary:**

This paper describes the INR-based video codec NVRC. NVRC is optimized E2E and includes a quantized model which is critical for device reproducibility (though this is not discussed in the paper). The performance on UVG is good and significantly better than VVC VTM 20.0. Benchmarks are given for an NVIDIA 4090 and shows near real-time for decode but 100X slower than real-time for decode. Source code will be released when accepted.

**Strengths:**

NVRC provides SOTA performance on the UVG dataset, provides a quantized model which is necessary for device reproducibility and efficient inference (both speed and power), and will release source code. Decoding time is near real-time on a NVIDIA 4090. It is a good solution for streaming scenarios.

**Weaknesses:**

The encoding time is 100X off from real-time, making it not approprate for any real-time scenarios (e.g., video conferencing, surveillance, etc). There is no discussion of device reproducibility. Is the entire system quantized or is there still some floating point operations that could break device reproducibility? The evaluation is only done on UVG which is insufficient. Others to include (see DCVC-DC paper) are: MCL-JCV, HEVC B, HEVC C, HEVC D, HEVC E, HEVC RGB.

**Questions:**

Is NVRC device reproducible, i.e., can the same stream be decoded accross any device? This would be a huge advantage. Floating point neural codecs are generally not device reproducible.

In order for NVRC to be practical for video streaming the decode needs to run on NPUs. What is needed to run on an NPU and what is the benchmark for that (say an Apple M3).

**Limitations:**

Yes, though because the evaluation is pretty weak (only UVG) there may be video scenarios that perform much worse than UVG.

---

> ### Author Rebuttal · Authors · 2024-08-07
>
> ***Q: For real-time scenarios***
>
> A: We agree with the reviewer that the proposed method is not yet appropriate for real-time applications. The relatively long encoding time is one of the main limitations of this AND other INR-based compression methods. We will mention this in the paper as an important topic for future work.
>
> ***Q: The reproducibility issue***
>
> A: Regarding reproducibility, we have not verified it for our work. As our model only utilizes a small convolutional network for autoregressive coding, and simple element-wise operations for network parameter coding, we believe that our network is compatible with integer operations and lookup table techniques, which enables reproducibility [Balle2018]. We will include a discussion of reproducibility when describing the limitations in our paper.
>
> ***Q: More databases for evaluation***
>
> A: We agree with the reviewer that performing evaluation on multiple datasets is important. Additional (preliminary) experiment results on the MCL-JCV and JVET CTC Class B datasets with the RGB settings are provided in the rebuttal (in the results pdf). Here we used the JVET CTC Class B rather then JCT-VC CTC Class B (HEVC-B) as the former is the latest HD testset for VVC. It can be observed that promising results have been observed in both dataset. The performance of NVRC is significantly better than HiNeRV, the previous best INR-based method in the MCL-JCV dataset. It is noted that the performance of NVRC with the MCL-JCV dataset is worse compare with the other datasets. We believe that this is mainly due to the shorter length of the sequences, and the above observation is consistent with previous work [24]. We will include full results in the camera-ready version if the paper is accepted.
>
> ***Refs***
>
> [Balle2018] Ballé, Johannes, Nick Johnston, and David Minnen. "Integer networks for data compression with latent-variable models.", ICLR. 2018.

---

> > ### Comment · Reviewer_YLxP · 2024-08-11
> > **Reply to rebuttal**
> >
> > Thanks for the response. I maintain my accept rating.

---

### Official Review · Reviewer_csHr · 2024-07-13

**Soundness:** 3
**Presentation:** 3
**Contribution:** 3
**Rating:** 7
**Confidence:** 4

**Summary:**

- This paper proposes an INR-based video codec, NVRC, which aims to improve the rate efficiency by encoding parameters hierarchically. Experimental results of the proposed method have been shown in RD performance on the UVG dataset.

**Strengths:**

- Experiments show good RD performance compared to recent INR-based codecs ([24], [23], [57]) (Figure 3), supporting this as an effective approach.

**Weaknesses:**

- I thought that the usefulness of the method could be better demonstrated by showing how much the RD performance was improved for each of the changes in the proposed method (1-4 on page. 2).
- The fact that the comparison was made with only one UVG dataset is also a weak point in demonstrating the general effectiveness of the method.

**Questions:**

- Have you compared each of the changes in the proposed method (1-4 on page. 2) with other methods ([24], [23], [57])? For example, V4 in Table 3 is intended to show the usefulness of hierarchical parameter coding, but have you compared it with the parameter coding method used in INR-based codecs ([24], [23], [57])?

**Limitations:**

- Section A.4 describes the limitations of the proposed method.

---

> ### Author Rebuttal · Authors · 2024-08-07
>
> ***Q: Showing coding gain for each contribution on Page 2.***
>
> A: Thank you for your suggestion. In the original paper, we presented these figures in the ablation study, including (contribution 2) the use of different quantization setting and entropy models (V1/V2/V3 for entropy models and V5 for quantization) and (contribution 3) the hierarchical parameter coding (V4).
>
> We are conducting additional ablation studies for V6 (pretraining + fine-tuning [13, 37, 57], i.e., without fully end-to-end optimized settings) and V7 (without alternating optimization) to confirm our contributions 1 and 4, respectively. Preliminary results for V6 have been included in the rebuttal pdf file.
>
> ***Q: More databases for evaluation***
>
> A: We agree with the reviewer that performing evaluation on multiple datasets is important. Additional (preliminary) experiment results on the MCL-JCV and JVET CTC Class B datasets with the RGB settings are provided in the rebuttal (in the pdf). Here we used the JVET CTC Class B rather then JCT-VC CTC Class B (HEVC-B), as the former is the latest HD testset for VVC. It can be observed that promising results have been observed in both dataset. The performance of NVRC is significantly better than HiNeRV, the previous best INR-based method. It is noted that the performance of NVRC with the MCL-JCV dataset is worse compare with the other datasets. We believe that this is mainly due to the shorter length of the sequences, and the above observation is consistent with previous work [24] We will include full results in the camera-ready version when the paper is accepted.
>
> ***Q: Compare each change with [24], [23], and [57]. ***
>
> A: Due to the large amount of experimentation required, we were unable at this stage to directly compare our approach with the parameter coding methods used in other INR-based codecs. However, the variants in our ablation study (including the additional results provided in the rebuttal) share some common and important features with these methods; the results indicate the effectiveness of our compression model. Specifically:
>
> - V6 (in the rebuttal results) employs the pre-training + fine-tuning settings used in [24, 57]. The results of this experiment show that the fully end-to-end optimized pipeline (in NVRC) does offer improved performance.
>
> - V5 uses a fixed quantization step size for feature grids, which is the same as that in [23]. The results demonstrate its inferiority compared to the learned step size. In addition, as mentioned in the paper (Line 332), we have applied the fixed quantization step size for neural network parameters, but the networks do not converge well.
>
> Furthermore, our proposed NVRC framework employs HiNeRV [24] as the neural representation model with minimal changes; we demonstrate significant performance gain compared to HiNeRV with the original compression pipeline (Table 1 in the paper). We believe that this is mainly due to our improved encoding and compression pipeline. We are happy to provide the results of the NVRC pipeline with the original HiNeRV in the final paper if needed.
>
> Compared to C3 [23], our model type is very different: in NVRC, HiNeRV is used as the neural representation model, where NeRV-style networks are much larger than the network used in C3. In C3, the neural network only contains a very small number of parameters, and the rate of the network is excluded in the training loop and the network is simply compressed with searched quantization parameters. This method is therefore inefficient in our setting.
>
> We will modify the text and include the justification text/additional results mentioned above in the revised paper.

---

> > ### Comment · Reviewer_csHr · 2024-08-11
> >
> > Thank you for your thoughtful reply.
> > I have increased my rate.

---

### Author Rebuttal · Authors · 2024-08-07

Thank you the reviewers for thorough feedback to our submission. We will address the concerns individually.

---

### Decision · Program_Chairs · 2024-09-25

**Decision:**

Accept (poster)

**Comment:**

The paper introduces NVRC, an INR-based video codec with reported excellent performance.

The authors provided satisfactory responses and convinced
- Reviewer csHr to increase the score to Accept
- Reviewer BGkd to increase the score to Borderline Accept
- Reviewer m68p to increase the score to Weak Accept
- Reviewer YLxP to maintain Accept

Thus, for this work there is an acceptance consensus.

After reading the paper and carefully checking the reviews and the authors' responses the ACs agree with the reviews that the present work makes important contributions and is of interest for the community.

The authors are invited to further refine their paper for the camera ready by including (part of) information/details from their responses to the reviewers' comments.